# A Single-Nucleus Transcriptomic Atlas of the Mouse Lumbar Spinal Cord: Functional Implications of Non-Coding RNAs

**DOI:** 10.3390/biotech14030070

**Published:** 2025-09-03

**Authors:** Pablo Ruiz-Amezcua, Miguel Nieto Hernández, Javier García Flores, Clara Plaza Alonso, David Reigada, Teresa Muñoz-Galdeano, Eva Vargas, Rodrigo M. Maza, Francisco J. Esteban, Manuel Nieto-Díaz

**Affiliations:** 1Molecular Neuroprotection Group, Instituto de Investigación Sanitaria de Castilla-La Mancha, National Hospital for Paraplegics (SESCAM), 45071 Toledo, Spainmiguelnietohernandez99@gmail.com (M.N.H.); javigarflo11@gmail.com (J.G.F.); plazaalonsoclara3a@gmail.com (C.P.A.); dreigada@sescam.jccm.es (D.R.); rodrigom@sescam.jccm.es (R.M.M.); 2Systems Biomedicine Unit, Department of Experimental Biology, University of Jaén, 23071 Jaén, Spain; evargas@ujaen.es

**Keywords:** spinal cord, single-nucleus RNA sequencing, neuronal subtypes, non-coding RNAs, transcriptomic atlas

## Abstract

The adult lumbar spinal cord plays a critical role in locomotor control and somatosensory integration, whose transcriptional architecture under physiological conditions has been characterized in various studies with restricted numbers of individuals (up to four). Here, we present an integrative single-nucleus RNA sequencing (snRNA-seq) atlas of the healthy adult mouse lumbar spinal cord, assembled from over 86,000 nuclei from 16 samples across five public datasets. Using a harmonized computational pipeline, we identify all major spinal cell lineages and resolve 17 transcriptionally distinct neuronal subtypes. A central novelty of our approach is the systematic inclusion of non-coding RNAs (ncRNAs), including long non-coding RNAs (lncRNAs) and pseudogenes. By comparing transcriptomic analyses based on coding-only, non-coding-only, and combined gene sets, we show that ncRNAs, despite accounting to a 10% of the recorded information of each cell, contribute to cell type-specific signatures. This resource offers a high-resolution, ncRNA-inclusive reference for the adult spinal cord and provides a foundation for future studies on spinal plasticity, injury, and regeneration.

## 1. Introduction

Recent advances in single-nucleus RNA sequencing (snRNA-seq) have significantly enhanced our understanding of cellular diversity in the spinal cord. This technology enables high-resolution profiling of gene expression at the level of individual nuclei circumventing limitations associated with dissociation and allowing access to fragile cell types. These features make snRNA-seq particularly valuable for studying central nervous system (CNS) tissues.

One of the most comprehensive efforts to date was conducted by [1], who generated an integrated reference atlas of the mouse spinal cord by harmonizing data across multiple studies, developmental stages, and anatomical regions. Notably, they introduced SeqSeek (https://seqseek.ninds.nih.gov; last accessed, 25 July 2025), an open-access platform that allows researchers to compare and classify transcriptomic profiles against a standardized reference. Subsequent studies by [2,3,4,5] focused on the lower thoracic- lumbar segments, applying snRNA-seq to characterize cellular responses to traumatic injury, rehabilitation, and epidural electrical stimulation. These efforts revealed both conserved and injury-induced transcriptional states, particularly in excitatory and inhibitory interneurons, and highlighted the complexity of spinal cell types under different physiological and pathological contexts. More recently, [6] employed single-nucleus and spatial transcriptomics to examine lesion-associated heterogeneity in the spinal cord. Their findings demonstrated context-dependent shifts in cellular composition and gene expression, underscoring the dynamic and region-specific architecture of spinal tissue. Notably, they leveraged all obtained data into *Tabulae Paralytica* (http://tabulaeparalytica.com; last accessed, 25 July 2025), a web-based platform that provides interactive insights into the gene expression changes following spinal cord injury.

Together, these studies provide a robust foundation for transcriptomic mapping of the spinal cord. However, the high cost of snRNA-seq has constrained previous studies to analyzing data from restricted numbers of individuals (typically fewer than four replicates per condition). [1] demonstrated the feasibility of data integration across studies, successfully combining samples from different ages, regions, and sequencing technologies. Building on this foundation, the availability of various publicly accessible snRNA-seq datasets now enables a unified analysis of over 80,000 nuclei from 19 individuals, specifically targeting the adult low-thoracic and lumbar spinal cord–a region critical for locomotion, proprioception, and lower-body reflexes. This level of coverage offers a unique opportunity to construct a region-specific atlas with greater depth and statistical power.

Additionally, the latest mouse genome assembly (GRCm39, June 2020) increased the number of annotated loci compared to GRCm38. GENCODE release M33 (July 2023), which is the annotation based on GRCm39 that we have employed in this study, comprises 21,403 protein-coding genes, 14,842 long non-coding RNAs (lncRNA) genes, 13,809 pseudogenes, and 6105 small non-coding RNA loci, corresponding to 149,488 transcripts in total (https://www.gencodegenes.org/mouse/stats_M33.html; last accessed, 20 August 2025). While most existing atlases focus on protein-coding genes, the mammalian genome is predominantly transcribed into ncRNAs–lncRNAs, antisense RNAs, and pseudogenes–that regulate diverse biological processes. In the CNS, ncRNAs are implicated in neural development, synaptic plasticity, cell-type specification, and injury response [7]. Although numerous studies have reported differential ncRNA expression following injury or in neurodegenerative models, their role in defining cell identity under baseline conditions remains largely unexplored.

Incorporating ncRNAs into a physiologically grounded, integrative atlas of the adult lumbar spinal cord provides a novel opportunity to uncover previously overlooked layers of transcriptional regulation. Unlike conventional atlases, our approach captures the full transcriptomic landscape by analyzing coding-only, non-coding-only, and combined gene sets, allowing us to assess the distinct contributions of ncRNAs to cell-type discrimination and clustering resolution.

Here, we present an integrated snRNA-seq atlas of the adult lumbar-low thoracic spinal cord, based on harmonized data from six studies. We developed a unified computational pipeline for quality control, doublet removal, normalization, batch correction, dimensionality reduction, clustering, and multimodal annotation. By explicitly incorporating non-coding gene expression into our analysis, we evaluate whether ncRNAs carry biologically meaningful signatures that refine our understanding of spinal cord cell types and functional organization.

Our results provide compelling evidence that non-coding RNAs contribute robust, subtype-specific signals that are not redundant with protein-coding profiles. In particular, certain ncRNAs display restricted expression in inhibitory, sensory, or neuropeptidergic populations, suggesting that they may act as lineage markers or regulatory switches. Moreover, by integrating codifying and non-codifying transcriptomes, we achieve enhanced resolution in neuronal subclusterization, improving the discrimination of functionally distinct subtypes.

## 2. Materials and Methods

### 2.1. Data Selection and Preprocessing

To construct a single-nucleus transcriptomic atlas of the adult mouse lumbar spinal cord under physiological conditions, we employed the pipeline outlined in Figure 1. This pipeline is mainly based on the procedures employed by [1] for preprocessing and integration of the samples and those of [6] for the clustering of the cells. All procedures employed publicly available code and software as indicated in Figure 1.

To select the samples to construct the atlas, we curated snRNA-seq datasets available until 2024 meeting these criteria: (1) origin from low thoracic-lumbar spinal cord, (2) healthy, adult, non-injured wildtype mice, (3) availability of raw FASTQ or processed count matrices, and (4) use of 10× Genomics Chromium technology without any prior sorting procedures (such as Fluorescence-activated cell or nuclei sorting). Appendix A summarizes sequencing depth, platforms, and metadata.

Each sample was processed independently to accommodate dataset-specific characteristics. Sequence quality of concatenated runs of each sample downloaded from SRA was assessed using FastQC (v0.12.1) [8]. Demultiplexing and alignment were carried out with STARsolo (v2.4.1) [9,10] using the Cumulus workflow (https://cumulus.readthedocs.io/en/latest/starsolo.html; last accessed, 19 November 2024) on TERRA Platform (https://app.terra.bio; last accessed, 17 December 2024). All samples were aligned to the same genomic reference built using GRCm39 mouse genome assembly and the GeneCode basic CHR gene annotation M33 (https://www.gencodegenes.org/mouse/release_M33.html; last accessed, 20 August 2025). Ambient RNA contamination and empty droplets were addressed with CellBender (v0.3.0) [11] with default False Discovery Rate (0.01) using the platform TERRA (Figure 1).

### 2.2. Quality Control and Cell Filtering

Count matrices (in H5 format) were imported into the R (v4.5.1 for Windows) [12] package Seurat v5.3.0 [13] for quality control (QC). Sample sex was determined when necessary (only [1] reported it specifically). Cells with ≥200 detected genes and genes present in ≥3 cells were retained. QC was performed independently per sample. Mitochondrial/ribosomal content and log10-transformed genes per unique molecular identifier (UMI) were used to identify low-quality nuclei. Doublets were detected using scDBlFinder (v1.13.4) [14] with 10 stochastic runs per sample; only consistently flagged doublets were removed. Cell cycle phase scores (S and G2M) were estimated using Seurat’s CellCycleScoring function and regressed out during data scaling to reduce confounding effects. The effectiveness of quality controls was confirmed by visual inspection of Uniform Manifold Approximation and Projection (UMAP) embeddings.

### 2.3. Normalization and Integration

Filtered samples were merged into a unified dataset. Normalization was performed using Seurat’s LogNormalize (v4.3.0), followed by HVG selection using the ‘vst’ method. Dimensionality reduction was performed with principal component analysis (PCA). Harmony integration (RunHarmony, 50 PCs) was applied to align datasets while preserving local structure [15]. Following prior studies [1,6], Harmony integration was selected for its performance and scalability in single-nucleus analyses.

### 2.4. Gene Set Definition: Coding, Non-Coding, and Combined

Filtered samples were merged into a unified dataset. Before subsequent analyses, three datasets were identified based on the information of Gene biotypes:

Combined (ALL): Dataset with all transcripts included in the M33 annotation of the genome;

Coding genes (CG): a subset of the transcripts annotated as “protein_coding”;

Non-coding genes (NCG): a subset of the transcripts annotated as “lncRNAs”, “antisense”, “pseudogenes”, “TECs”, and “non-coding isoforms”.

### 2.5. Clustering

To comprehensively resolve the cellular diversity of the adult mouse spinal cord, we implemented a two-step clustering strategy:

(i) Coarse clustering of major spinal cord cell types. To define the major cellular compartments of the mouse spinal cord, we performed low-resolution clustering on the three datasets: CG, NCG, and ALL. For each subset, the top 30 Harmony-corrected principal components, selected based on visual inspection of the ElbowPlot, were used to construct a k-nearest neighbor (k-NN) graph with k = 20. Clustering was then performed using the Leiden algorithm with Seurat’s default implementation [16]. To ensure comparability across datasets, we explored clustering at multiple resolutions using Clustree (v2.10.0) [17] and selected the resolution that produced a consistent number of clusters across all subsets: 11 clusters each for the ALL (resolution = 0.07), CG (resolution = 0.07), and NCG (resolution = 0.2) datasets.

(ii) Fine subclustering of neuronal populations. To further resolve the transcriptional diversity of spinal neurons, we performed a dedicated subclustering analysis starting from the coarse-level clusters obtained in the initial integration. Instead of extracting neuronal cells from the original count matrices, as performed by [1], we applied multiple rounds of clustering directly on the fully integrated atlas, following the approach proposed by [6]. This strategy involves performing successive clustering at increasingly fine resolutions, enabling the identification of both major cell types and more granular neuronal subtypes.

### 2.6. Cell Type Annotation

Annotation combined manual and automated approaches. AddModuleScore function from Seurat was used to compute cell type-specific gene program scores using [1] marker sets. Alternative automated annotation was performed using SingleR [18] with the *Tabulae Paralytica* reference (available at http://tabulaeparalytica.com/). To manually confirm specific annotations, marker genes (via FindAllMarkers) were cross-referenced with SeqSeek, *Tabulae Paralytica*, and literature.

### 2.7. Compositional Analysis

Number of cells per cluster was compared among studies, sexes, and chemistries using scCODA (v0.1.9) [19] to identify potential effects of these covariates on cluster abundances. scCODA is a Bayesian framework specifically designed for compositional single-cell data that models the relative proportions of cell types while considering inherent negative correlative bias via joint modeling of all measured cell type proportions. The method estimates credible effects by comparing each covariate against a reference cell type automatically selected by the mode, thereby providing robust inference on whether observed abundance differences reflect true biological or technical effects rather than proportional artifacts. Each factor (Chemistry, Study, and Sex) was defined as a covariate and analyzed independently. We set the reference cell type to be automatically identified by the model. The script and all outputs for each dataset are available at OSF (https://osf.io/xwfsz/; last accessed, 28 August 2025) and in Appendix A.

### 2.8. Data Analysis

Comparisons among groups were carried out using *t*-test, ANOVA or Kruskal–Wallis test for non-parametric data. In comparing annotations, we used Simpson and Shannon diversity indexes [20] to quantify the dispersion of populations among clusters. For each cluster, Simpson diversity index was calculated as 1 − Σ(n(n − 1)/N(N − 1)), where ‘n’ is the number of cells from a population and ‘N’ is the total number of cells, whereas Shannon diversity index was calculated as H = −Σ(pᵢ × ln(pᵢ)), where pᵢ is the proportion of cells belonging to population i and ln indicates the natural logarithm. Data is expressed as mean ± standard deviation unless specified. Statistical analyses were carried out using R or JASP [21].

Scripts and detailed procedures are provided in Appendix A and at OSF (https://osf.io/dbgxt/; last accessed, 28 August 2025 and https://osf.io/xwfsz/; last accessed, 28 August 2025). Editing support was provided by the Scientific Writing Assistant [22].

## 3. Results

### 3.1. Quality Control

19 samples of naïve, wildtype, adult C57BL/6 mouse low thoracic-lumbar spinal cords from six studies were identified (Table 1). All samples were originally processed with 10x Genomics and the sequencing data and associated metadata are available at the NCBI’s GEO and SRA repositories (see also Appendix A). Inspection of sequencing data with FastQC platform did not reveal any relevant quality issue (each sample report is available at OSF: https://osf.io/dbgxt/). To prevent pseudoreplication, the three samples from the GSE165003 study [4] were excluded from the analysis because they were equal to samples in the GSE184370 study [3]. All other samples were demultiplexed and aligned to the mouse genome assembly (GRCm39) and the full annotation M33 (basic annotation (CRH) M33) covering 21,403 coding genes and 35,481 non-coding genes and pseudogenes.

The resulting count matrices for each sample were subjected to quality controls to exclude background signal noise, empty droplets, cell doublets and compromised cells using CellBender and various functions from Seurat (Figure 2). In general, all samples consisted predominantly of high-quality cells, with doublets accounting for 3–8% of the events (10–14% in [2]). In the [1] dataset, three of the four samples (GSM4798623, GSM4798624 and GSM4798626) displayed a defined cluster of cells with high mitochondrial content and low complexity. These clusters comprised 147, 223 and 99 cells, respectively. By contrast, the remaining sample did not show a clear cluster, retaining only residual proportions of cells with these features. In [5] dataset, sample GSM5961591 exhibited a cluster of 206 cells with high mitochondrial content. In [6], all three samples contained very few cells with low complexity and high ribosomal content. In addition, two of them (GSM7474501 and GSM7474503) showed small clusters with high mitochondrial content, comprising 142 and 64 cells, respectively. Following QC assessment, anomalous cells were discarded–including removal of cells expressing fewer than 200 genes and genes detected in fewer than three cells.

### 3.2. Clustering of Spinal Cell Types

The filtered nuclei from each sample were merged and integrated into three different datasets containing all transcriptomics information (ALL), or subsets with only information on coding (CG) or non-coding genes (NCG). The number of nuclei included remained the same in the three datasets (86,378). Median gene counts per nucleus were 1430.2 for the coding dataset and 144.2 for the non-coding, summing 1574.6 in the combined dataset. The three datasets were processed independently to obtain a coarse clusterization of the cells for each type of data.

We employed the Leiden clustering method on 30 PCA components from each subset to identify coarse clusters segregating the main spine cell types. According to Clustree (see Appendix A), NCG dataset showed lower cluster resolution than CG and ALL datasets; therefore, we employed 0.07 resolution for CG and ALL datasets and 0.2 for NCG dataset to cluster the cells into 11 groups in the three datasets (Figure 3).

### 3.3. Annotation of Major Clusters

To assign cell types to each identified cluster we employed the automated annotations obtained using SingleR with *Tabulae Paralytica* reference (comprising 92 cell populations) and AddModuleScore with the SeqSeek cell markers (for 81 cell populations). Comparison of the automated annotations revealed clear differences between the two reference atlases. Inspection of the distribution of the populations from both references within the clusters identified here revealed that most populations identified according to *Tabulae* annotations were restricted to one cluster (ALL dataset, Simpson diversity index = 0.96; Shannon index = 0.10), whereas the populations identified using SeqSeek markers were more distributed among multiple clusters, leading to a higher proportion of mixed or ambiguous assignments (ALL dataset, Simpson diversity index = 0.58; Shannon index = 0.84; both *p* < 0.01 relative to *Tabulae* values).

Based on these observations, we concluded that the *Tabulae Paralytica* + SingleR annotations provided greater consistency for our dataset and therefore adopted them for subsequent analyses.

### 3.4. Comparison of Clusterings Among ALL, CG, and NCG Datasets

According to the annotation from *Tabulae Paralytica*, the 11 clusters obtained from each of the three datasets consistently corresponded to the major spinal cord cell lineages with some interesting differences. Both the complete (ALL) and the protein coding (CG) datasets yielded the same clusters (Table 2), namely, one cluster of mature and differentiating oligodendrocytes and one cluster of oligodendrocyte precursor cells (OPCs), one cluster of astrocytes, one cluster of vascular cells, one cluster of microglia and other immune cells, one cluster of ependymal cells, and five clusters of neurons. Neuronal clustering at this coarse level separated three clusters of different dorsal horn excitatory neurons, one cluster of dorsal horn inhibitory neurons expressing galanin neuropeptide (DI_Gal in *Tabulae Paralytica* terms), and a large cluster containing all other types of neurons, including a few populations of dorsal horn excitatory neurons, most dorsal horn inhibitory neurons, and all ventral neurons, including motoneurons.

Non-coding information gave rise to a clustering pattern that in broad terms and in some clusters–such as those of astrocytes and ependymal cells–is highly consistent with the ALL and CG clustering patterns. However, the clustering of NCG dataset yields three oligodendrocyte clusters, two of mature and differentiating cells and one of OPCs. This supposed the split of the ALL and CG mature oligodendrocyte clusters into two clusters (numbers 1 and 6), the second one containing half of the mature oligodendrocytes annotated as ischemic whereas the other half is included in Cluster 1. A second major difference concerns neurons, which are split in only three clusters, a large one (Cluster 2) with all types of neurons except dorsal horn excitatory neurons, that cluster together in a second group (Cluster 3, all dorsal excitatory (DE) except for DE Maf-Rorb-Cpne4). The third cluster (number 10) is composed of ventral neurons from three populations mainly represented in Cluster 2. A third major difference concerns immune cells, which appear divided into peripheral and central (microglial) populations, the former one associated with vascular cells. There is also a small cluster, corresponding to cells from sample GSM5243303, without consistent assignment to any population. Despite these differences, as a whole, the clustering patterns from the three datasets are mainly consistent with one another, differentiating among the major neural cell lineages, the vascular, and the immune cells. The distribution of the 83,000 cells within clusters across datasets is similar in both the number of cells in each major cluster and in the cells that form part of the cluster, irrespectively of the employed datasets (Table 2, Figure 4). The markers (coding and non-coding genes) for each cluster obtained after processing the ALL, CG, and NCG datasets are available in Appendix A.

Clustering patterns are also similar to those obtained in previous studies [2,4,6]. They are highly similar to the 14 clusters present in layer 2 of *Tabulae Paralytica*, including an early split in the oligodendrocyte lineage among OPCs and mature and differentiating oligodendrocytes, and the loose grouping of astrocytes and ependymal cells. However, the clustering patterns clearly differ in neurons, which in *Tabulae* appears as a single cluster that is divided into dorsal and ventral populations at higher resolution, whereas in our analyses appear dispersed in up to five clusters, with a major division between DE neurons and the rest of neuronal populations. Relative to SeqSeek atlas [1] and the studies that employed it [2], the basic grouping remain consistent but with some differences, particularly in the presence of meningeal and Schwann cell clusters and a consistent neuronal cluster that it is not supported by our results.

### 3.5. Cluster Composition Is Retained Among Samples

One objective of this study is to test whether the composition, in terms of cell typologies and abundances, is preserved between samples, with their specific methodological conditions (Table 1). A general inspection of the cluster abundance in the different samples (Figure 5) reveals that all clusters are present in all samples with consistent abundances. Lack of consistency is observed in the clustering obtained using the NCG dataset, where cluster 11 is exclusively composed of cells from sample GSE5243303 and cluster 10 is absent in four samples.

To evaluate the consistency across datasets, we calculated the coefficient of variation (CV) of the abundance of cells in each cluster across the samples of each dataset. As shown in Figure 5D, CVs are low to moderate for most clusters (<0.5), whereas those corresponding to vascular, immune, and ependymal cells show higher variations (>0.5). CV values are in the range described by [23] for five individuals in his stereological analysis of the spinal cord cell composition (range: 0.109 to 0.750 depending on the cell and location). The employed dataset does not seem to deeply affect the consistency among samples (H Kruskal–Wallis = 0.962, *p* = 0.612).

To further explore the potential effects of the Study, Chemistry or Sex we employed scCODA, a Bayesian modeling specifically designed to perform compositional data analysis in scRNA-seq. Analyses in the three datasets confirm that Sex does not affect the cellular composition of any of the clusters. On the contrary, both the Chromium chemistry version and the Study affected the abundance of several clusters. All scripts, results and graphs are available at Appendix A.

The analysis of the effects of the study indicates that the samples included in study GSE198949 [4] have a credible effect (according to scCODA terminology) on the abundance of the astrocyte and the large neuronal clusters in the three datasets (ALL, CG, and NCG), as well as on the immune cluster in ALL and CG datasets. This is in agreement with our visual observations on the anomalous abundances in the clusters of samples GSM5961586 and GSM5961588, both from GSE198949 study (Figure 5). These two samples contain the lowest number of reads and number of cells in any of the samples and are included in an analysis devoted to analyze the relative performance of different methods of differential expression and their ability to account for variation between biological replicates in which the three included undamaged spinal cord samples show strong differences in the number of reads and cells. Additional effects were observed on the astrocyte cluster in study GSE184370. The scCODA analyses also revealed effects of the Chromium version on the same astrocyte, vascular, neuronal, and immune cell clusters in the three datasets. Major effects are identified in the Chemistry vs. 3, which was employed in study GSE198949.

A further analysis on the effects of the number of reads, chemistry and study on the properties of the datasets revealed that, despite not affecting their cellular composition, the samples processed using the 10x Genomics Chromium version 2 have fewer number of counts and genes per cell than the samples processed with later versions (Figure 5, see also Appendix A). This effect was independent of the employed dataset (ALL, CG, or NCG). Interestingly, it was also independent of the sequencing depth measured as the number of reads per cell, indicative of reaching saturation at fairly low sequencing depths. On the contrary, most samples analyzed using versions 3 and 3.1 showed a linear relationship of the information of each cell and the sequencing depth, indicating that they were far from saturation. However, at the present low resolution clustering, these effects did not affect the clustering patterns, not even in the NCG dataset, despite the low number of counts and genes detected in each cell.

Overall, leaving out the data from the anomalous samples GSM5961586 and GSM5961588, the resulting values (Table 3) indicate that the low thoracic to lumbar spinal cord is composed mainly of glial cells (median percentage: 53.9%, with oligodendrocytes contributing to a 41.1% of the total spinal cells), whereas neurons account for nearly one third (median 36.5%), with immune and endothelial cells contributing together to a 10%, and non-neuronal to neuronal (nNNR) and glial to neuron (GNR) ratios of 1.8 and 1.6, respectively. Data also reveal that these abundances are consistent across samples, studies, chemistries, sequencing depths, or even the type of data employed (coding or non-coding).

### 3.6. Fine Clustering and Neuronal Populations

To achieve finer cell clustering, we analyzed the ALL dataset, covering both coding and non-coding genes, at a higher resolution of 1 (Clustree graph available in Appendix A). The analysis identified 35 clusters, nine corresponding to oligodendrocyte and OPCs, three to the astro-ependymal lineage, two to immune cells, three to endothelial cells, one is a mixed cluster, and the remaining 17 corresponding to neuronal populations (see Figure 6A and Appendix A). Focusing on the neuronal clusters, we can identify seven clusters of dorsal excitatory neurons that comprise all cells within the 11 DE populations established in *Tabulae Paralytica*. Three of these populations (DE_Tac2_Nmu, DE_Cck_Cpne4, and DE_Reln_Trhr) appear in their own clusters, isolated from the other populations, whereas the remaining DE populations appear in clusters comprising 2 or more DE populations. Only the population DE_Rreb1_Zim1 appears distributed among different clusters including a mixed cluster comprising ventral and medial neurons. Some of the clusters combining two or more populations agree with the hierarchical clustering available at *Tabulae Paralytica* such as cluster 18, which comprises DE_Maf_Cpne4_Rorb and DE_Maf_Kcnh8 cells, two populations closely related in *Tabulae Paralytica*. The other group of spinal neurons that appear clearly defined are the dorsal inhibitory (DI) neurons, which are distributed into five clusters. Two of them correspond to single populations from the *Tabulae Paralytica*, namely DI_Gal and DI_Npy_Qrfpr. DI_Gal neurons appear clearly separated from the rest of neurons even at coarse resolution. The remaining eight DI populations described in *Tabulae Paralytica* are distributed in the remaining three DI clusters, except for the DI_Npy_Vgf population that appears in a mixed neuron cluster.

Contrary to the dorsal neurons, the ventral and medial neurons are not resolved in this analysis. These neurons are distributed within five clusters that combine multiple populations, in many cases overlapping with other clusters. A compositional analysis using scCODA indicates that the three largest of these clusters are influenced by the sample of origin (Figure 6B; Appendix A), suggesting that the transcriptional differences among ventral and medial neuron populations are small, lesser than the transcriptional differences derived from the methodological differences between studies or samples. Compositional analyses also identified effects of the study or the employed chemistry in other four neuronal clusters, namely cluster 4, composed of a combination of all kind of neurons, including all CSF-contacting neurons (CSF-cN) in the sample, and clusters 14, 19, and 23, which are clusters of various DE and DI populations. It is interesting that only the study GSE198949 [4] was identified to affect the neuronal clusters, which was the study that also identified to affect the composition of the coarse clusters, likely due to the inclusion of two anomalous samples. Trying to resolve these mixed populations and to reduce the effects of the study and chemistry on the clustering, we also performed a dedicated subclustering analysis on all cells labeled as “neurons” in the coarse clustering, following [1] pipeline (the clusters of neuronal cells from each sample were merged and integrated using Harmony before clustering). However, the resulting clusters were even more heavily influenced by the study design and sample chemistry, with many clusters showing no clear association with cellular populations. We also attempted to subcluster the neuronal cells from the ALL dataset, with even worse results. Cluster composition was well conserved among samples, with only minor deviations. All neuronal and non-neuronal clusters identified at higher resolution were consistently detected, except for cluster 35 (203 cells), which was absent in two samples. Cell-type abundance was also preserved, as indicated by the coefficient of variation (CV) of cluster sizes across samples (Figure 6C). Most clusters displayed low to moderate variability (CV < 0.5). Higher variability (CV > 0.7) was observed in eight clusters, particularly among immune and vascular populations (clusters 11, 24, 34, and 35) and in heterogeneous mixed clusters (clusters 10 and 17). Notably, anomalous values were concentrated in samples GSM5961586 and GSM5961588, consistent with patterns observed in the coarse-resolution clusters. Excluding these two samples markedly reduced the CVs, resulting in strong agreement in the relative proportions of the 35 clusters across the remaining 14 samples (Figure 6C).

### 3.7. Non-Coding RNA Markers of Spinal Populations

Differential expression analyses were conducted using the FindAllMarkers function within our integrated Seurat dataset to identify the markers of each cluster. Our analysis primarily highlighted the presence of non-coding RNAs, including pseudogenes and long non-coding RNAs, demonstrating significant cell-type specificity.

Our current findings, detailed in Appendix A, reinforce the fundamental role of ncRNAs in defining cell subtype identities across various lineages. Specifically, the oligodendrocyte lineage exhibited a substantial presence of pseudogenes and lncRNAs, particularly within clusters 1 through 9. Markers such as *Gm42413*, *6030407O03Rik*, and *Gm37459* suggest potential roles in differentiation and specific oligodendrocyte states, including ischemic conditions (clusters 4 and 9).

Neuronal populations showed significant diversity in ncRNA markers, with pseudogenes and lncRNAs widely distributed across clusters (Figure 7). Notably, Cluster 17 (“mixed neurons”) exhibited an exceptionally diverse ncRNA profile, including microRNAs (miRNAs; e.g., Mir6236), long intergenic non-coding RNAs (lincRNAs; e.g., *Meg3*), and ribosomal RNAs (rRNAs; e.g., *Rn18s-rs5*). It is worth mentioning that, according to scCODA compositional analyses, there are differences in the composition of this cluster depending on the version of 10x Chromium employed. Moreover, a closer look at the composition of this cluster indicates that it is particularly enriched on cells from sample GSM7474503 of [6] study.

Astroglial clusters (3, 27, and 30) and immune-related clusters (11 and 34) also presented pseudogenes and lncRNAs as distinctive markers. Vascular endothelial cluster 35 uniquely featured an antisense lncRNA (*Tbx3os1*), indicating specific regulatory functions. Cluster 10, characterized by mixed vascular, endothelial, and neural stem cell populations, featured a singular mitochondrial ribosomal RNA (*mt-Rnr1*), potentially reflecting specialized metabolic or regulatory states. In contrast, clusters 13 (OPCs), 24 (arachnoid leptomeningeal cells), 26 (endothelial), and 32 (neuronal DI_Npy_Qrfpr) lacked notable ncRNA markers.

## 4. Discussion

The findings presented herein provide a comprehensive and high-resolution view of the cellular architecture of the adult mouse lumbar spinal cord under physiological conditions. By integrating over 86,000 nuclei from five public datasets, we constructed a single-nucleus transcriptomic atlas that identifies all major spinal cord cell lineages and resolves 17 transcriptionally distinct neuronal subtypes. This broad coverage and analytical depth complement previous studies such as those by [1,6], by systematically incorporating non-coding information and enhancing statistical power through the integration of a larger number of nuclei.

In constructing this atlas, we also identified technical and curatorial challenges inherent to the integration of datasets from independent sources. For instance, we excluded the samples from GSE165003 [4] because they were exact duplicates of those in GSE184370–an issue not clearly annotated in public repositories, which could have introduced pseudoreplication artifacts. Additionally, we observed that in a few samples–all from [1]–there were cells with low complexity and high mitochondrial content that tended to co-cluster, suggesting a shared underlying low-quality phenomenon likely related to partial RNA degradation. These findings reinforce the importance of thorough quality control beyond standard filters, particularly when working with archived datasets, and the convenience of carrying out these analyses in each individual sample before integration. A final technical challenge we have analyzed here concerns the annotation of a specific region, such as the lumbar spinal cord from adult mice. We evaluated two different approaches–SingleR with *Tabulae Paralytica* reference and AddModuleScore with SeqSeek markers–observing strong differences in granularity and consistency. While *Tabulae*-based annotations tended to assign each cluster to a dominant and coherent cell type, SeqSeek-derived annotations resulted in a more fragmented distribution, with many clusters containing apparent mixtures of cell types. Importantly, the discrepancies should not be attributed to the annotation method, as annotation using SingleR with the SeqSeek reference yielded similarly mixed clusters such as those observed using AddModuleScore. Instead, they most likely reflect fundamental differences between the two reference atlases. *Tabulae Paralytica* was derived from a large dataset of spinal cord samples from uninjured (used here) and injured adult lumbar cords, all processed using the same snRNA-seq and Drop-seq (10x Genomics Chromium) platforms employed in our datasets. In contrast, SeqSeek was constructed from 52,623 cells across six datasets of spinal cord samples spanning early postnatal, juvenile, and adult mice, and processed using both single-cell and single-nucleus approaches, with Drop-sex, Split-seq, and Fluidigm methodologies [1]. This methodological and developmental heterogeneity likely underlies the inconsistent annotations and highlights the key importance of the reference in cell annotation.

Inclusion and relevance of non-coding RNAs

A key methodological innovation of this study was the systematic inclusion of ncRNAs–including lncRNAs, pseudogenes, antisense transcripts, lincRNAs, and ribosomal RNAs–in clustering and differential expression analyses. According to our analyses, non-coding transcripts constituted 10% of the detected transcripts (most samples contain less than 150 non-coding genes and 300 counts per cell). Despite the scarcity of non-coding data, it was able to group the cells under study in clusters compatible with those obtained using the coding genes. Moreover, this approach uncovered transcriptional patterns that were not detectable when analyzing coding genes alone. That is the case of the split of the oligodendrocytes into two clusters (numbers 1 and 6) in the coarse clustering of the NCG dataset, which was not observed in the analyses of the ALL and CG datasets. A comparison of the markers of both clusters showed that, although several are shared between the two populations (6 out of 10 in cluster 1 and 5 out of 10 in cluster 6) and many remain poorly characterized, the remaining markers revealed key differences and valuable insights. Long non-coding RNAs associated with oligodendrocytes, such as *Neat1*, *Sox2ot*, and *4930419G24Rik* ([28,29,30]; see also mousebrain.org), were consistently identified as markers of the larger cluster 1, but not of the smaller cluster 6. Conversely, the lncRNA *miR100HG*–a primary microRNA transcript encoding microRNAs *let-7a-2*, *miR-125b*, and *miR-100*–was specifically enriched in cluster 6. Previous studies have proposed that this pri-miRNA may contribute to determining the fate of OPCs toward astrocytes rather than to oligodendrocytes [31]. Moreover, the progressive downregulation of the *Mir100HG* embedded microRNAs characterize the differentiation of the oligodendrocyte lineage [32]. Together, non-coding genes suggest that the small cluster 6 may correspond to a population of non-fully committed oligodendrocytes that was ignored when employing coding gene information alone. However, further analyses are necessary to confirm this hypothesis.

Other non-coding elements such as *6030407O03Rik*, *Gm42413*, *Gm37459*, *Mir6236*, *Meg3*, and *Tbx3os1* exhibited highly restricted expression across specific cell populations, suggesting their potential roles as lineage markers or as functional elements in transcriptional regulatory networks. In this context, the biological roles of most of the ncRNA markers identified in our atlas remain largely unexplored, but existing literature provides plausible hypotheses for their functional relevance in the CNS. For instance, *Meg3* has been pinpointed as a lncRNA involved in synaptic plasticity in neurons, one of the main mechanisms supporting the cognitive functions of the brain (learning, memory, adaptation to stimuli, and protection against neuronal damage). More specifically, *Meg3* seems to be involved in the regulation of the PTEN/PI3K/AKT signaling cascade [33]. *Meg3* also functions as an anti-inflammatory regulator in the spinal cord, as its overexpression inhibits microglial M1 polarization and attenuates neuroinflammation after acute spinal cord injury [34]. We hypothesize that the restricted expression of *Meg3* in neuronal populations detected in our study may contribute to activity-dependent regulation of excitatory circuits. Further, *Tbx3os1* has been identified as a pericyte-related marker that may be linked to senescence in a single-cell RNA sequencing study of the aged mouse brain [35]. Notably, the specific expression of *Tbx3os1* in endothelial cells may reflect specialized regulatory functions within the spinal vascular microenvironment. While these functional interpretations remain speculative, they underscore the potential of ncRNAs as regulators of lineage-specific transcriptional networks and highlight promising candidates for future experimental validation in the context of spinal cord physiology and pathology.

Despite the lower gene counts per nucleus in the non-coding dataset, we found that this information retained sufficient transcriptional variability to allow effective clustering. Interestingly, when comparing the clustering structures derived from coding, non-coding, and combined data, we observed that the inclusion of non-coding genes contributed to a more hierarchical and consistent partitioning of major spinal cell types. This suggests that non-coding RNAs enhance the robustness of transcriptomic classification, not only by increasing resolution at the subtype level but also by stabilizing broader lineage-level distinctions across analytical resolutions.

In addition to expanding the resolution of major lineages, our analysis highlights the underexplored yet critical role of non-coding RNAs as molecular markers in the spinal cord. The absence of these ncRNAs in conventional marker lists based on coding genes underscores their value as complementary classifiers, particularly for distinguishing fine subtypes within the oligodendrogial and neuronal lineages. The diversity of ncRNA markers–ranging from pseudogenes and lincRNAs to microRNAs and antisense transcripts–observed in specific neuronal and glial clusters suggests the presence of complex regulatory networks. This is especially evident in mixed neuronal clusters, where ncRNAs may reflect distinct physiological states or technical artifacts related to read depth and chemistry, as suggested by the enrichment of certain clusters in samples with low coverage. Conversely, the absence of ncRNA markers in specific clusters may indicate a predominant reliance on protein-coding or other regulatory transcripts for subtype specification. Together, these findings reinforce the importance of systematically including non-coding elements in transcriptomic atlases and downstream functional studies. Our results emphasize the importance of using updated reference annotations comprising both coding and non-coding genes, as well as selecting appropriate alignment tools in the generation of count matrices from sequencing data.

Feasibility of integrated snRNA-seq for spinal cord cell type composition and abundance analysis

A central outcome of our study is the demonstration that integration of snRNA-seq datasets from multiple studies provides a reliable means to analyze the cellular composition and abundance of spinal cord populations. Previous single-nucleus studies of the spinal cord [2,3,4,5,6] have provided important insights into cell type diversity and responses to injure but have generally been based on three samples per condition. Such limited datasets are insufficient to capture inter-individual variability. Importantly, our results confirm that this approach is applicable not only to broad, multi-organ studies but also to anatomically specific regions such as the lumbar spinal cord, which is of particular relevance for locomotor and sensory functions.

Integration of single-nucleus transcriptomics not only resolves transcriptional identities but also recapitulates the proportional representation of neurons, oligodendrocytes, astrocytes, microglia, and vascular cells. As shown in Table 3, our results for the low thoracic to lumbar spinal cord indicate a composition dominated by glial cells, with a non-neuronal to neuronal ratio (nNNR) of 1.8 and a glial to neuron ratio (GNR) of 1.6. Both values are consistent with previous stereological analyses by [23] and snRNAseq studies such as [6]. In contrast, these findings differ significantly from those of [26], who reported an nNNR of 9.9 and a GNR of 7.3. This discrepancy is likely due to their use of whole cells instead of nuclei, which introduces a bias against cells with complex morphologies like neurons and oligodendrocytes. We also noted a lack of consistency in the results from [27], who initially reported an nNNR of 0.9 and a GNR of 0.5. However, a subsequent reanalysis of their data by [1] yielded an nNNR of 2.6 and a GNR of 1.5, the latter being very similar to our value of 1.6. The most significant differences are seen when comparing our results with the isotropic fractionator estimates from [24,25], which suggest a much higher proportion of non-neuronal cells (nNNR between 3.2 and 4.1). These discrepancies warrant a detailed analysis to understand their underlying causes, which may include the specific region of the spinal cord sampled, the age and sex of the individuals, or the inherent biases of each quantification method.

Comparison with previous atlases also highlights notable similarities and distinctions in the abundances of major cell types. In general, the estimated cellular proportions–predominantly glial (~54%), followed by neurons (~36%) and immune-vascular cells (~10%)–are consistent to a certain extreme to the stereological data from [23] and the snRNAseq data from [27]. A higher agreement is observed to the abundances described by [6] in their *Tabulae Paralytica*, logical considering their data constitutes a significant part of the data here analyzed and the reference for cell annotation. However, unlike the *Tabulae Paralytica* resource, in which dorsal and ventral neuronal populations are grouped together at low resolution, our analysis disaggregates these populations into multiple functionally defined clusters. 

Data also reveal that these abundances are highly consistent across samples, studies, chemistries, sequencing depths, or even the type of data employed (coding or non-coding). Analysis of the compositional consistency across samples revealed an overall preservation of the proportions of the major cell types as well as of the more specific populations, with low-to-moderate variability across most clusters. However, two samples from the GSE198949 study (GSM5961586 and GSM5961588) displayed anomalous cluster abundances and low sequencing depth, which likely influenced their divergence in cell composition. To avoid biases in cell-type representation, we excluded these outlier samples from the final integration. The application of scCODA confirmed credible effects of both the study and the Chromium chemistry version on the abundance of several astrocytic, neuronal, vascular, and immune clusters, reinforcing the need to account for technical covariates in multi-study analyses. Explicitly, its Bayesian framework allowed us to distinguish credible technical effects from genuine biological variation, thus increasing confidence in the robustness of the observed cell-type abundances. 

Overall, the observed robustness of the cluster proportions across samples underscores the reproducibility of snRNA-seq-based compositional analyses, even when integrating data obtained under heterogeneous technical and experimental conditions. It also validates data integration as a reliable strategy for generating biologically meaningful inferences about the relative abundance of spinal cord cell populations. This establishes snRNA-seq as a dual-purpose tool: one that can capture molecular diversity at the single-cell level while simultaneously providing robust information about the relative abundance of major and minor cell populations.

After excluding the outliers, the estimated cell-type proportions aligned closely with stereological benchmarks [23] and with recent single-cell studies such as [6], indicating a predominance of glial cells (~54%) followed by neurons (~36%) and a minor fraction of vascular and immune cells (~10%). These estimates differ substantially from earlier atlases such as [26,27], whose neuron counts appear, respectively, underestimated and overestimated–discrepancies that were partly resolved in the reanalyses conducted by [1]. This supports the reliability of the current integration framework and highlights the value of rigorous QC and statistical modeling in establishing accurate cell composition references.

Resolution of neuronal populations and technical limitations

While clustering patterns were broadly consistent with those reported in previous atlases such as *Tabulae Paralytica*, our results diverged in the organization of neuronal populations. In contrast to the coarse dorsal-ventral division in *Tabulae* or the unified neuronal cluster observed in some earlier studies [36], our initial low resolution analysis identified up to five transcriptionally distinct neuronal clusters, including a specific separation between dorsal excitatory neurons and all other neuronal subtypes.

Our fine-resolution analysis further confirmed that several dorsal excitatory and inhibitory neuronal subtypes formed well-defined and transcriptionally distinct clusters, in some cases mirroring the hierarchical structure described in *Tabulae Paralytica*. In contrast, ventral and medial neurons showed greater heterogeneity and lower resolution, appearing scattered across multiple overlapping clusters.

Compositional analysis using scCODA revealed credible effects of study or origin and the Chromium version on the abundance of ventral and medial neurons. These findings suggest that the transcriptional differences among these ventral subtypes were smaller than the variance introduced by methodological factors such as study origin or sequencing chemistry. This was especially evident in clusters influenced by the GSE198949 dataset. Attempts to resolve these populations through additional subclustering approaches–including the direct reclustering of the neuron subset–proved ineffective, likely due to residual batch effects and low intrinsic variability between subtypes. These findings emphasize both the strengths and current limitations of single-nucleus transcriptomics for resolving complex neuronal diversity, particularly when integrating data across multiple studies.

Altogether, this atlas represents a valuable reference resource for neurobiological research, offering a novel integration of the non-coding transcriptome within the central nervous system. Beyond providing a robust tool for the physiological characterization of the spinal cord, this work lays a solid foundation for future investigations into synaptic plasticity, spinal cord injury responses, and the identification of therapeutic targets grounded in transcriptional and epigenetic regulation. Future studies should aim to functionally assess the most specific ncRNAs and explore their dynamic expression in models of spinal cord injury and regeneration.

Limitations and caveats

The integration pipeline effectively reduced technical variability across studies; however, important limitations persist. Residual methodological differences–such as the version of the 10x Genomics sequencing chemistry–were evident in specific clusters associated with outlier samples, which were excluded from quantitative analyses. Although batch correction mitigated inter-study variation, differences in sequencing depth and chemistry version may still influence the detection of low-abundance ncRNAs, which may contribute to variability across clusters and should be considered when interpreting these results. Furthermore, the study lacks functional validation (e.g., in situ hybridization, perturbation assays) to confirm the cell-type specificity of the identified ncRNAs. Finally, the atlas is restricted to healthy adult tissue, limiting insight into the dynamics of ncRNA signatures during development or in pathological states. In addition, our current clustering approach did not resolve transcriptionally distinct subtypes within ventral neuronal populations, which remained dispersed across overlapping clusters, likely influenced by the study or chemistry of origin. These technical (batch) effects may be compensated through interaction methods. Here, we employed Harmony given its broad standardization and its balance between performance and scalability. However, other methods such as reciprocal PCA (RPCA) may prove useful in future works. In summary, all these events may reflect insufficient transcriptional variability, batch effects, or limitations inherent to the resolution of single-nucleus data in specific neuronal lineages.

## Figures and Tables

**Figure 1 biotech-14-00070-f001:**
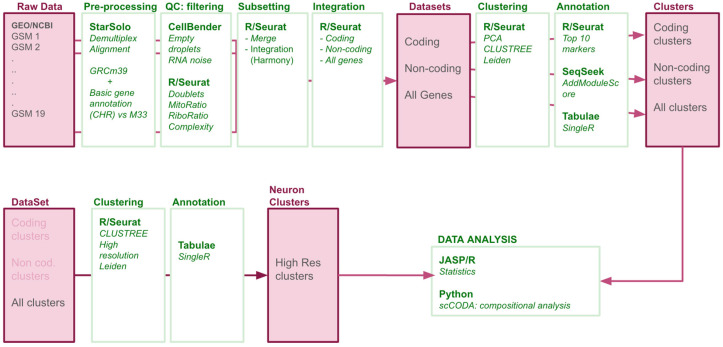
Data Processing Pipeline. Following download from Gene Expression Omnibus (GEO), concatenation of the different reads per sample, and the quality controls of the sequences, each resulting FastQ file was aligned to the GRCm39 mouse genome with M33 annotation, including coding and non-coding genes. Post-alignment, we performed rigorous quality controls to remove gene- and cell-level noise and artifacts. Samples were then merged to produce three datasets: coding-only, non-coding only, and combined. Each dataset was independently integrated, followed by Leiden clustering and manual annotation to compare major clusters. The combined dataset was further analyzed to identify discrete neuronal subpopulations using the same clustering strategy applied to neuron-bearing clusters.

**Figure 2 biotech-14-00070-f002:**
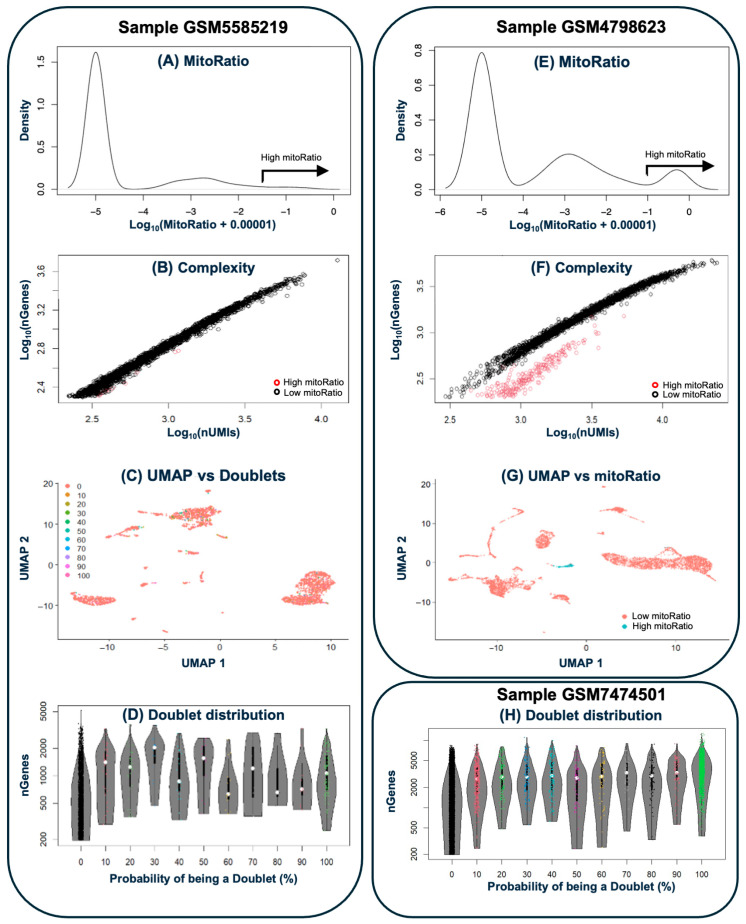
Quality control. Left side shows (from top to bottom): (**A**) the distribution of cells by mitochondrial gene content, (**B**) a complexity plot (UMIs vs. genes), (**C**) a UMAP visualization of cell distribution, and (**D**) the distribution of predicted doublets for sample GSM5585219), which is representative of samples without significant quality control issues. (**E**–**G**) depicts sample GSM4798623, which presents a population of anomalous cells characterized by high mitochondrial gene expression and low complexity. (**H**) shows doublet distribution in sample GSM7474501, which presents the highest percentage of doublets in all analyzed samples (13.2%).

**Figure 3 biotech-14-00070-f003:**
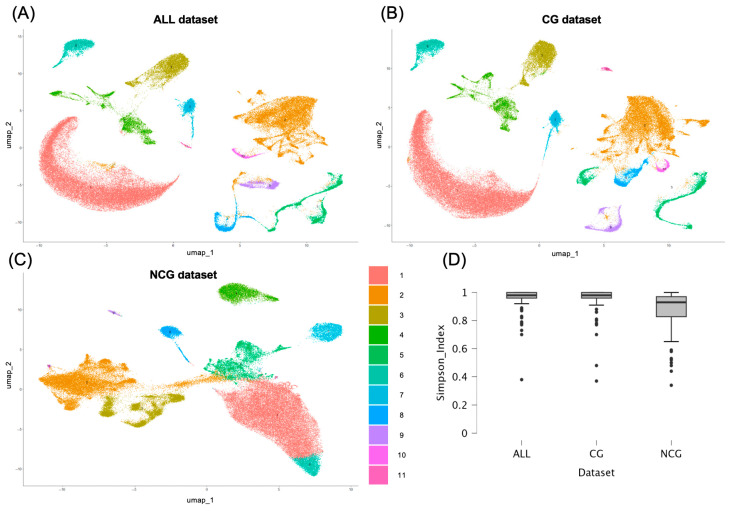
Coarse clustering. UMAPs for ALL (**A**), coding (**B**), and non-coding (**C**) datasets. Cluster colors and codes in the legend apply to all UMAPs. (**D**) Boxplots comparing the distribution of cell populations annotated through single R and *Tabulae Paralytica* reference among the clusters identified with the ALL, CG, and NCG datasets. The Simpson diversity index was used to assess whether each annotated population was restricted to specific clusters (values close to 1) or more broadly distributed (values closer to 0).

**Figure 4 biotech-14-00070-f004:**
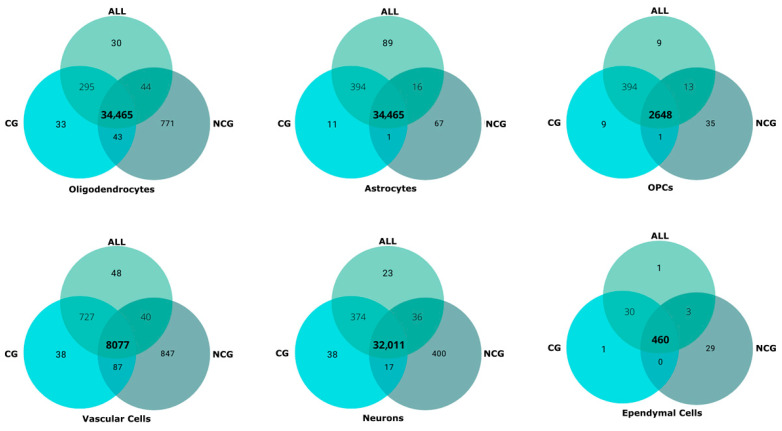
Comparison of the cells included in the clusters from coding + non-coding (ALL), coding (CG), and non-coding (NCG) datasets. Venn diagrams illustrate the shared cells among the major clusters. OPCs: oligodendrocyte precursor cells.

**Figure 5 biotech-14-00070-f005:**
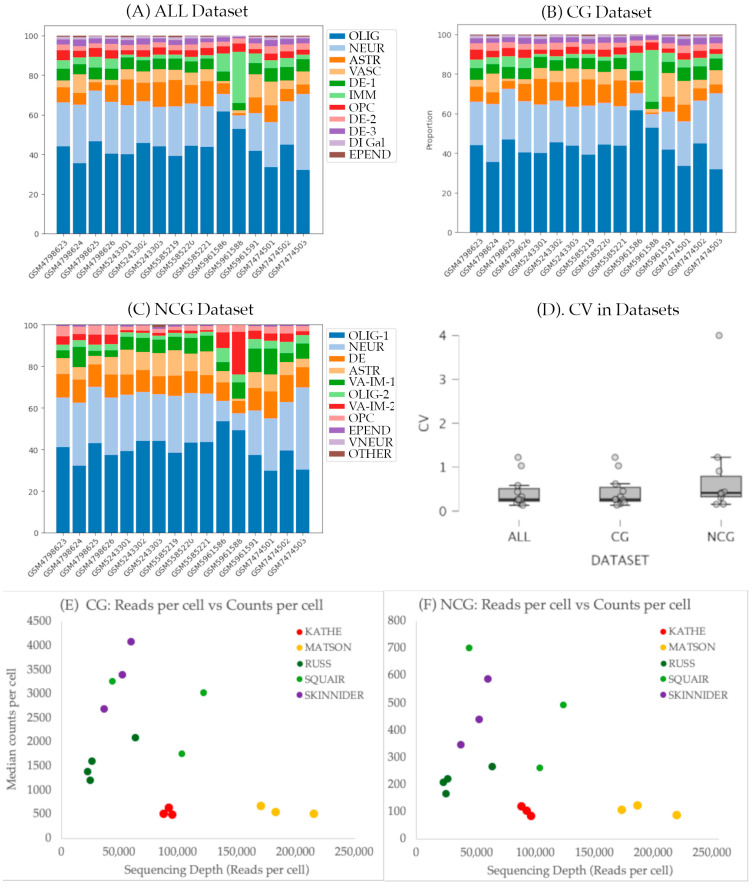
(**A**–**C**) Bar diagrams showing cellular abundance in each cluster in the three datasets: ALL (**A**), coding (**B**), and non-coding (**C**) datasets, respectively. (**D**) Boxplot showing the distribution of the coefficient of variation (CV) in the abundance of each cluster in all samples of each dataset; ALL: coding + non-coding; CG: coding; NCG: non-coding; OLIG: oligodendrocytes; NEUR: neurons; ASTR: astrocytes; VASC: vascular cells; DE: dorsal excitatory neurons; IMM: immune cells; OPC: oligodendrocyte precursor cells; DI-Gal: dorsal inhibitory neurons expressing galanin; EPEND: ependymal cells; VA-IM: mixed cluster of vascular and immune cells; VNEUR: ventral and medial neurons. (**E**,**F**) Effects of the number of Reads and the Chromium version on the counts per cell. Reads were normalized to the total number of cells of the CG (**E**) and NCG (**F**) datasets. Chemistry (version 2, 3 o 3.1) are shown in different colors (Chemistry 2 in red-yellow, 3 in green, and 3.1 in purple). Data and additional graphs are available in Appendix A.

**Figure 6 biotech-14-00070-f006:**
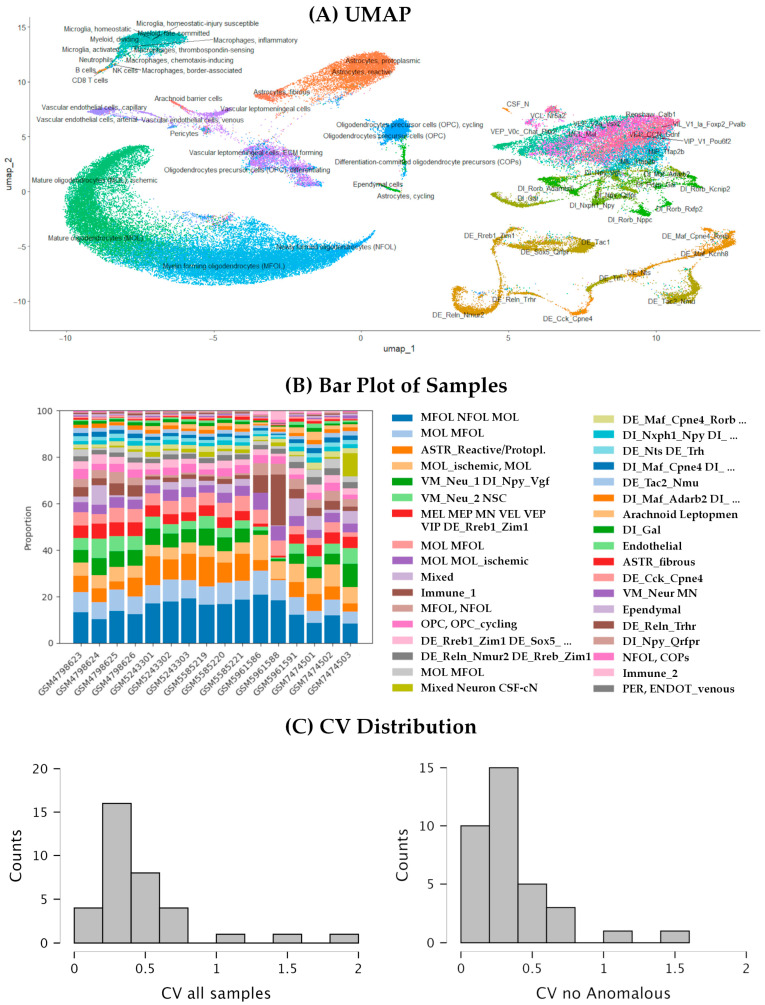
Fine clustering analysis and relationships between neuronal populations in the integrated spinal cord atlas. (**A**) UMAP representation of the integrated dataset showing automatic cell type annotation using SingleR and the *Tabulae Paralytica* reference. (**B**) Bar plot showing the proportion of cells from each study within each cluster. (**C**) Histogram of the coefficient of variation (CV) in the abundance of each population across samples, when considering all samples (left) and without the anomalous samples GSM5961586 and GSM5961588. The detailed composition of each cluster is provided in Appendix A.

**Figure 7 biotech-14-00070-f007:**
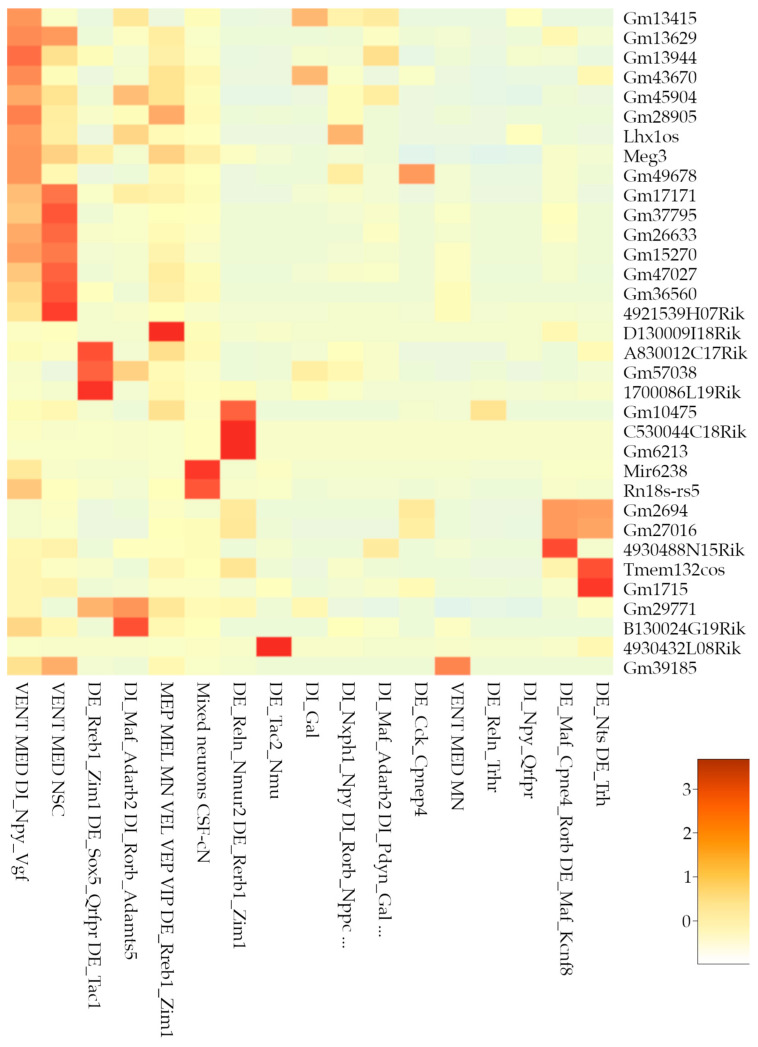
Non-coding markers of neuronal clusters. Heatmap showing the non-coding markers for each neuronal cluster in the high-resolution analysis. Legend shows z-score values.

**Table 1 biotech-14-00070-t001:** Samples included in the study.

GEO Series ID	10x Genomics Chromium Version	GEO Samples ID (Number of Runs in SRA)	Reads (M)	Sex	Age (Weeks)	Publication
GSE158380	Single Cell 3’ Kit Version 3	GSM4798623 (6)	172.4	M	9	[1]
		GSM4798624 (6)	166.0	M		
		GSM4798625 (6)	174.2	F		
		GSM4798626 (6)	178.7	F		
GSE165003	Single Cell Kit Version 2	GSM5024317 (8) *	267.2	F	12–30	[4]
		GSM5024318 (8) *	282.8	F		
		GSM5024319 (8) *	274.5	F		
GSE198949	Single Cell 3’ Kit Version 3	GSM5961586 (1)	84.1	F	8–15	[5]
		GSM5961588 (2)	100.6	F		
		GSM5961591 (8)	903.2	F		
GSE172167	Single Cell Kit Version 2	GSM5243301 (15)	505.6	F	12–30	[2]
		GSM5243302 (15)	593.9	F		
		GSM5243303 (15)	648	F		
GSE184370	Single Cell Kit Version 2	GSM5585219 (8)	267.2	F	12–30	[3]
		GSM5585220 (8)	282.8	F		
		GSM5585221 (8)	274.5	F		
GSE234774	Single Cell Kit Version 3.1	GSM7474501 (8)	698.2	F	8	[6]
		GSM7474502 (8)	465	F		
		GSM7474503 (8)	480.2	F		

* Samples in dataset GSE165003 were excluded due to repetition with samples in dataset GSE184370. M: male; F: female. Additional data is available at Appendix A.

**Table 2 biotech-14-00070-t002:** Coarse clustering of spinal cord cells using coding, non-coding, and complete (ALL) datasets.

Lineage	Coding Genes	Non-Coding Genes	All Genes
Oligodendrocytes	1 (34,836) 98.9%	2 (32,302/3021) 97.6%	1 (34,834) 98.9%
Oligodendrocyte precursor cells	1 (2968) 89.2%	1 (2697) 98.2%	1 (2986) 88.7%
Astrocytes	1 (6635) 93.9%	1 (6313) 98.7%	1 (6728) 92.6%
Ependymal	1 (491) 93.7%	1 (492) 93.5%	1 (494) 93.1%
Vascular	1 (5603) 89.7% *	2 (6158/2893) 89.2%	1 (5485) 90.8%
Microglia/immune	1 (3405)		1 (3407)
Neurons	5 (32,440) 98.7%	3 (32,464) 98.6%	5 (32,444) 98.7%
VE, VI, DI	1 (21,168)	1 (22,903)	1 (21,396)
DE	3 (5141/2601/2542)	1 (9455)	3 (5179/2573/2307)
DI-Gal	1 (988)		1 (989)
VE-VI		1 (106)	
Other		1 (38 **)	
Total clusters	11 (86,378)		

Cluster annotation based on the automatic annotation using SingleR and *Tabulae Paralytica* reference. For each cell type, the number of included clusters, the number of cells in each cluster (between brackets), and the % of cells in common with equivalent clusters in the other 2 datasets are indicated. * indicates a % estimated by joining vascular and immune data to make it comparable to the clusters resulting from NCG dataset. ** corresponds to cluster 11 of NCG dataset comprising 38 cells of sample GSM5243303. VE: ventral excitatory; VI: ventral inhibitory; DI: dorsal inhibitory; DE: dorsal excitatory; DI-Gal: dorsal inhibitory expressing galanin neuropeptide.

**Table 3 biotech-14-00070-t003:** Abundance of major cell types in the ALL dataset and comparison with previously published abundance data.

Source	Oligodendrocytes	Neurons	Astrocytes	Vascular Cells	Immune Cells	OPCs	Ependymal Cells	Other Cells	nN/N	GNR	Methodology
ALL Dataset (Median)	43.0	35.5	9.0	6.0	2.5	4.0	1.0		1.8	1.6	snT
ALL Dataset (Mean)	41.1	36.5	8.9	6.1	3.1	3.4	0.6		1.7	1.5	snT
[24]									3.7		IF
[25]—4 weeks									3.2		IF
[25]—40 weeks									4.1		IF
[23]	39.3	33.6		19.1				8.0	2.0	1.2	STE
[26] (in [1])	40.4	9.2	18.0	18.6	5.2	6.4	2.0	0.1	9.9	7.3	snT
[6]	49.7	35.3	3.4	4.2	3.1	3.3	0.7		1.8	1.6	snT
[27] (in [1])	24.8	28.0	13.6	7.9	2.4	1.6	2.4	19.3	2.6	1.5	snT
[27]	16.0	52.0	9.0	5.0	1.0	1.0		14.0	0.9	0.5	snT

For abundance calculations, values from samples GSM5961586 and GSM5961588 were excluded. OPCs: oligodendrocyte precursor cells; snT: single nucleus transcriptomics; IF: isotropic fractionator; STE: stereology; nN/N: non-neuronal to neuron ratio; GNR: glial to neuron ratio. Data on abundance from [26,27] were obtained from the harmonized analysis by [1], in addition to the values from the original analysis in the case of [27]. Abundance of cell types is expressed as % of the total number of cells.

## Data Availability

Scripts and detailed procedures are provided at https://osf.io/dbgxt/ and https://osf.io/xwfsz/.

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
