# Peer review of "A Single-Nucleus Transcriptomic Atlas of the Mouse Lumbar Spinal Cord: Functional Implications of Non-Coding RNAs"

_biotech, 2025, doi:10.3390/biotech14030070_

Round 1
Reviewer 1 Report
Comments and Suggestions for Authors
This study integrates coding and non-coding RNA data from multiple datasets to build a high-resolution adult mouse lumbar spinal cord atlas. It identifies novel ncRNA markers and refines neuronal subtype resolution, though ventral populations remain unresolved. Findings highlight ncRNAs’ value but require functional validation and improved batch effect mitigation. But, study needs some revisions for better understanding and readability.
Comments:
- The integration of non-coding RNA (ncRNA) expression into a spinal cord atlas is novel and potentially impactful. However, the functional roles of the identified ncRNA markers remain speculative. The discussion would benefit from stronger linkage to existing literature and explicit hypotheses on their biological roles.
- The removal of duplicate and outlier samples is justified, but the manuscript should clarify how these exclusions affect representation of rare populations.
- The authors should discuss whether alternative integration methods (e.g., SCTransform regression, reciprocal PCA) were explored in ventral/medial neurons part.
- The manuscript should discuss the impact of sequencing depth and chemistry on ncRNA detection sensitivity.
- The preference for Tabulae Paralytica over SeqSeek is reasonable, but a deeper comparison of the references’ differences in annotation resolution would strengthen the rationale of the study.
- The identification of cell type specific ncRNAs is a strength, yet functional validation is lacking. In silico approaches such as co-expression network or pathway enrichment analyses could support the biological relevance of these markers, if possible.
- Figures (e.g., clustering UMAPs) are dense and would benefit from zoomed panels or simplified layouts for key findings.
- The Bayesian framework of scCODA is appropriate but should be briefly explained for general readers.
Reviewer 2 Report
Comments and Suggestions for Authors
The authors present an integrated snRNA atlas of the healthy mouse lumbar spinal cord with an updated computational pipeline. Overall, the study is well demonstrated, and the methods are all described in detail. The only suggestion is that, for the cell atlas annotation based on ncRNA, besides the Sup Table 5 with more information, a heatmap showing the relative expression levels of the representative ncRNAs in the clusters of different cell types would greatly emphasize the diversity of ncRNAs and their potential for future study.
Reviewer 3 Report
Comments and Suggestions for Authors
The submitted research article reports a single-nucleus transcriptomic atlas of the mouse lumbar spinal cord by means data assembly from over 86,000 nuclei across five public datasets; the study combines coding-only, non-coding-only, and coding + non coding gene sets, with particular attention to non coding Rna. The manuscript is wll organized and stuctured. Data collection, integration and interpretation is fine; the limitations of the study are reported and discussed. This reviewer only has has few queries related to data presentation:
Figure 2: label panels and change the figure legend accordingly; enlarge the size of the font in panels (quite unreadeble in most panels at printing size)
Figure 3 and Figure 5: enlarge the size of the font in panels (quite unreadeble in A-C at printing size).
Figure 4 and Figure 6: enlarge the size of the font in panels (quite unreadeble in all panels at printing size)
Supplementary figure 1 has poor quality and it is quirte unreadeble
The title of Table 2, Table 3 and Supplementary table 2 contains the description table content and the definition of abbreviations; move these information in table legend
Use Italic font for the names or genes/RNA
